# Significance of Th17 and Treg in Treatment Efficacy and Outcome in Pediatric Acute Lymphoblastic Leukemia

**DOI:** 10.3390/ijms241512323

**Published:** 2023-08-01

**Authors:** Anna Krętowska-Grunwald, Małgorzata Sawicka-Żukowska, Małgorzata Kowalska, Aleksandra Basaj, Maryna Krawczuk-Rybak, Marcin Moniuszko, Kamil Grubczak

**Affiliations:** 1Department of Pediatric Oncology and Hematology, Medical University of Białystok, 15-274 Bialystok, Poland; malgorzata.sawicka-zukowska@umb.edu.pl (M.S.-Ż.); maryna.krawczuk-rybak@umb.edu.pl (M.K.-R.); 2Department of Regenerative Medicine and Immune Regulation, Medical University of Białystok, 15-269 Białystok, Poland; marcin.moniuszko@umb.edu.pl; 3Department of Pediatric Surgery and Urology, Medical University of Białystok, 15-274 Bialystok, Poland; gosiak0115@gmail.com; 41st Chair and Department of Cardiology, University Clinical Center of the Medical University of Warsaw, Central Clinical Hospital, 02-097 Warsaw, Poland; aleksandra.izabela.basaj@gmail.com; 5Department of Allergology and Internal Medicine, Medical University of Białystok, 15-276 Białystok, Poland

**Keywords:** ALL, Th17, Treg, ALL-IC BFM 2009, cancer immunity, pediatric cancer

## Abstract

Acute lymphoblastic leukemia represents a malignant proliferation of lymphoid cells blocked at an early stage of cell differentiation. It is the most common cancer occurring in children. Despite favorable prognosis, the survival rate of patients with poor treatment response or relapse remains dismal. The interaction between leukemic cells and the tumor immune microenvironment is pivotal in mediating tumor progression. In this study we evaluated associations between Treg and Th17 lymphocytes and the clinical presentation of ALL pediatric patients to validate their value in monitoring treatment outcome. The peripheral blood and bone marrow aspirates from 35 pediatric patients with ALL and 48 healthy control subjects were selected for the experiment. We demonstrated the numbers of Th17 lymphocytes and Tregs were increased in the bone marrow of ALL patients at the moment of diagnosis compared to the healthy control group, with the latter significantly decreasing during the course of ALL treatment. Patients with lower Th17 were found to demonstrate higher risk of blasts prevalence in bone marrow at day 33. ALL patients with lower WBC demonstrated higher frequency of Tregs. In summary, we identified a significant role of Th17 and Treg lymphocytes in ALL of pediatric patients and their contribution to disease-related parameters.

## 1. Introduction

Acute lymphoblastic leukemia (ALL) is the most commonly diagnosed childhood cancer, accounting for nearly a quarter of all malignant neoplastic diseases in the pediatric population. ALL is defined as a malignant proliferation of lymphoid cells blocked at an early stage of cell differentiation, with a dominance of B-cell lineage cases [1,2]. Due to significant advances in treatment of childhood acute lymphoblastic leukemia over the last decades, the overall survival rates of pediatric patients with ALL improved substantially, reaching 90% in patients with good prognosis [3,4]. Despite favorable efficiency of treatment, the prognosis of the remaining 10% who have relapsed or do not show sufficient response to therapy remains dismal. In addition, due to the prevalence of pediatric ALL, the associated deaths are the second most common cause of cancer-related deaths among children [5]. According to numerous studies, the tumor microenvironment, consisting of the majority of T-cells, macrophages and natural killer cells, seems to play a crucial role in the development of leukemia acting through impairment of the immune system [6,7].

Regulatory T cells (Tregs) are crucial elements in the maintenance of immunological homeostasis. Through specialized receptors (CTLA-4 (cytotoxic T cell antigen 4), GITR (glucocorticoid-induced TNFR-related protein)) and secreted immunosuppressive cytokines (interleukin 10 (IL-10), IL-35, transforming growth factor beta (TGF-ß)) Tregs regulate the excessive proinflammatory response of the immune system. In accordance, disturbed function of these cells results in autoimmune reactions [8,9]. Although the presence of Tregs is essential in preventing autoimmunity, they were shown to exert unfavorable influence within the tumor microenvironment. Immunosuppressive activity of elevated peripheral blood regulatory T cells of cancer patients was associated with a worse clinical outcome [10]. Noteworthy, reports demonstrated an increase in Tregs population also in pediatric patients diagnosed with hematological malignancies, including acute leukemia [11,12]. In addition, in acute myeloid leukemia (AML) patients’ changes in Tregs were not limited to their levels but also their function reflected by higher production of cytokines: IL-10, IL-35, TGF-ß [13].

Regulatory T cells are not the only immune cell population that can act as a double-edged sword in our immunity. IL-17-producing T cells (Th17) serve as inflammation promotors, inter alia in bacterial and fungal infection, releasing IL-17, IL-21, IL-22 and TNFα. However, this unique CD4+ T cell subset was also found to be linked to the pathogenesis of autoimmune conditions and chronic graft-versus-host disease [14,15,16]. Similar to Tregs, effector Th17 cells were also found to participate in the course of acute leukemia, including B-ALL pathogenesis. Patients with limited numbers of Th17 cells exhibited significantly lower survival rates [17]. Expression of Th17 lymphocytes was found to be elevated in patients with ALL prior to starting chemotherapy with their number decreasing throughout treatment [11]. Furthermore IL-17A in the peripheral blood was under investigation of causing potential resistance of B-ALL cells to daunorubicin through activation of Akt signaling, making Th17 cells a possible novel target for development of future B-cell ALL immunotherapy [18].

Reports on the involvement of Th17 and Treg populations in the immune responses associated with ALL justify great importance of the studies aimed at this area. Noteworthy however, to date, a scarce number of research is focused on a comprehensive combining of immunological profiles with clinically important aspects of ALL management, especially in the pediatric population. Therefore, here we aimed to evaluate the direct associations between selected T cell populations and the clinical presentation of our ALL patients to validate their value in monitoring therapy outcome.

## 2. Results

### 2.1. Increased Contribution of T17 and Treg Lymphocytes in the Course of Acute Lymphoblastic Leukemia in Pediatric Patients

Initial ex vivo analyses of the bone marrow revealed a 48% higher frequency of IL-17-producing T cells in acute lymphoblastic leukemia patients at the moment of diagnosis compared to the healthy control group (1.28% versus 0.86%; *p* = 0.0155). Similarly, regulatory T cells also demonstrated an increase in frequency of 89% in ALL subjects (3.13% versus 1.65%; *p* = 0.0134). Similar direction of changes did not affect the ratio between studied populations of T cells (*p* = 0.6913). Concomitant analyses of the peripheral blood also showed increased values of Th17 and Treg lymphocytes in ALL subjects. However, obtained data did not achieve statistical significance for studied populations: Th17 (*p* = 0.4109), Treg cells (*p* = 0.1139), and their ratio (*p* = 0.9328). In an additional evaluation, we again did not report differences in Th17 or Treg cells when comparing bone marrow and peripheral blood in both leukemic patients (Th17 *p* = 0.2285; Treg *p* = 0.6080) and the control group (Th17 *p* = 0.5703; Treg = 0.9999). Analysis of mutual correlations between studied immune cell populations revealed a significant link only between regulatory T cells of bone marrow and peripheral blood (r = 1.000; *p* < 0.0001) (Figure 1).

### 2.2. Association between Th17 and Treg Lymphocytes and ALL Laboratory Parameters Related to the Disease Course

Considering significant variations in Th17 and Treg cells between acute lymphoblastic leukemia patients at the moment of diagnosis and the control group, we proceeded with assessment of their correlation with diagnostic laboratory data. In reference to bone marrow immune cells, Th17 demonstrated moderate correlation with bone marrow and peripheral blood blasts frequency. In addition, negative association of these cells was exhibited in the context of the number of platelets in the blood. In the context of Th17 cells analyzed in peripheral blood, this population correlated positively with blood lymphoblasts, and negatively with platelets and hemoglobin to a moderate degree. Additional moderate association was also found in reference to blood Treg and leukocyte count in the blood. Despite the observations above, the statistical significance for correlation analyses was not reached (Figure 2).

Although no significant correlations were found above, we further investigated whether levels of presented laboratory parameters are characterized by any differences in selected immune cell populations. In fact, ALL patients with lower leukocyte count demonstrated higher frequency of regulatory T cells (*p* = 0.0390). The same tendency was found in the context of peripheral blood lymphoblasts where higher Tregs levels were detected in patients with lower lymphoblasts (*p* = 0.0311). None of the immune cell populations revealed any link with lower/higher levels of bone marrow lymphoblasts (Figure 3). Furthermore, regulatory T cells were found to have elevated frequencies in ALL patients with higher levels of platelets (*p* = 0.0073) and hemoglobin (*p* = 0.0056). In contrast to bone marrow results, we did not find significant association between Th17 or Treg frequencies and levels of the routine diagnostic laboratory data. The low/high laboratory parameters groups were based on the median levels of hematological parameters within studied patients: WBC (median = 6900 cells/μL), Blasts PB (median = 23%), Blasts BM (median = 88%), PLT (median = 95,000 counts/μL), and HGB (median = 7.7 g/dL).

### 2.3. Therapy Induced Changes in IL-17-Producing Lymphocytes and Regulatory T Cells of Acute Lymphoblastic Leukemia Patients; Th17 and Treg Cells Influence on the Treatment Course

Subsequently, we focused on the verification of therapy effects on the frequency of Th17 and Treg cells in pediatric acute lymphoblastic leukemia patients. First, no changes were demonstrated in reference to IL-17-producing T cells and treatment application. Significant differences, however, were found in Treg lymphocytes in the bone marrow with a decline reported on the 15th day of therapy (T1) (*p* = 0.0339). A reduced level of T cells with a regulatory phenotype was maintained through the 33rd day (T2; *p* = 0.0191) and after 2 months of anti-cancer treatment (T3; *p* = 0.0582). No statistically significant differences were shown in context of peripheral blood immune cells in the course of therapy. However, after 33 days of therapy, there seemed to be a gradual increase in regulatory T cell values (Figure 4A).

Considering the data presented above, we decided to analyze whether Th17, Treg or their ratio have a substantial effect on the response of ALL patients to the therapy applied. In accordance, we divided ALL patients into low/high groups on the basis of the subjects’ median values of Th17 (median = 0.91%), Treg (median = 2.04%), and Th17/Treg ratio (median = 0.73%). Regarding Th17 lymphocytes, patients with higher frequencies demonstrated lower levels of leukocyte count in blood (4.494 cells/μL; *p* = 0.0004) compared to low-level patients (11.110 cells/μL; *p* = 0.0290) on the 33rd day of therapy (T2). Interestingly, ALL subjects with higher Th17 counts had initially increased peripheral blood (42% versus 32%; *p* = 0.0766) and bone marrow (90% versus 81%; *p* = 0.0360) blasts compared to patients with low values of these immune cells. In addition, in the course of therapy, both groups of patients managed to obtain the same significant reduction of the lymphoblasts’ levels in studied compartments (*p* < 0.0001) (Figure 4B). Such differences between lower/higher level groups were not observed in context of Treg cells. However, peripheral blood and bone marrow blasts seemed to be slightly higher in those patients with low Treg levels. Moreover, ALL subjects with lower Tregs were not able to achieve a significant drop of leukocyte count towards more favorable values, in contrast to the high regulatory T cells group (T1 *p* = 0.0007; T2 *p* = 0.0043) (Figure 4C). Evaluation of the effects of Th17/Treg ratio levels on the therapy-induced responses revealed tendencies comparable to these reported in Th17 lymphocytes. High ratios were associated with higher initial blast levels in the peripheral blood (45% versus 31%; *p* = 0.0428) and bone marrow (91% versus 83%; *p* = 0.0670). Still, both groups achieved a beneficial decline in blast levels in the course of treatment (*p* < 0.0001). Furthermore, leukocyte count on the 33rd day (T2) was higher than expected in patients with low Th17/Treg ratio (30.666 cells/μL versus 17.564 cells/μL; *p* = 0.0106) (Figure 4D).

### 2.4. Evaluation of Th17 and Treg Lymphocyte Value in the Bone Marrow before ALL Treatment as Risk Predictors of Acute Lymphoblastic Leukemia Outcome in Pediatric Patients

Finally, to validate the prospective usefulness of Th17, Treg or Th17/Treg ratio implementation in the prediction of acute lymphoblastic leukemia therapy outcome, we performed a risk analysis assessment. None of the immune-based parameters allowed us to estimate the prevalence of death occurrence before treatment implementation. However, patients with lower Th17/Treg ratio at the time of diagnosis had significantly reduced chances of relapse, with a relative risk value of 0.08 (*p* = 0.0278). We did not demonstrate any prognostic potential of Th17 and/or Tregs for achievement of more beneficial reduction of bone marrow blasts at day 15 of therapy. Furthermore, patients with lower Th17 levels were found to demonstrate a higher risk of substantial blasts prevalence in bone marrow on day 33 (relative risk value = 3.66; *p* = 0.0363). In contrast, ALL patients with lower Th17 and Th17/Treg ratios were shown to demonstrate a lower risk of a presence of positive minimal residual disease (MRD) on day 15, with a relative risk value of 0.74 (*p* = 0.0206) and 0.75 (*p* = 0.0244), respectively. The patient group with lower regulatory T cells was found to have increased risk of demonstrating higher MRD values on day 15 of treatment (relative risk value = 1.41; *p* = 0.0500) (Figure 5).

### 2.5. Assessment of Proinflammatory Cytokines IL-10 and IL-17 in the Bone Marrow, in the Course of ALL Induction Therapy

Considering the substantial role of Th17 and Treg in the pediatric patients with ALL, finally, we analyzed bone marrow concentrations of the related cytokines: IL-17 and IL-10, respectively. Firstly, no differences were shown between studied patients and the control group at the admission stage in reference to IL-10 (*p* = 0.4814), IL-17 (*p* = 0.8208) and IL-17/IL-10 ratio (*p* = 0.3730) (Figure 6A). Interestingly, before therapy, ALL patients showed significant dominance of the IL-10 over IL-17 levels (*p* = 0.0071). The therapeutic approach diminished those differences with a significant reduction of IL-10, starting on the 8th day (*p* = 0.0026) and was maintained at a comparable level until the 33rd day of treatment (*p* = 0.0106). As IL-17 was also slightly reduced compared to the before therapy level (*p* = 0.0423), on the 33rd day of therapy we were again able to observe the difference between IL-10 and IL-17 concentrations; yet it was less pronounced (*p* = 0.0401) (Figure 6B). Subsequent analysis revealed that in leukemic patients the associations between Treg versus IL-10 (r = 0.80; *p* = 0.0051), and Th17 versus IL-17 (r = 0.86; *p* = 0.0143), are significantly stronger compared to the control group—Treg/IL-10 (r = 0.51; *p* = 0.0015) and Th17/IL-17 (r = 0.32; *p* = 0.0599). However, we observed that unlike healthy controls, ALL subjects did not show correlations between Th17 and IL-10 or Treg and IL-17 (Figure 6C). Linear regression analysis, with slope assessment included, allowed us to confirm significant differences between ALL patients and the control group in reference to mutual interactions between IL-10 and IL-17 (*p* = 0.0013) (Figure 6D).

## 3. Discussion

Successful advances in the treatment of pediatric acute lymphoblastic leukemia have dramatically improved the rates of positive outcomes in the majority of patients. Despite the overall survival reaching 90%, the prognosis for the remaining subset of non-responding patients remains poor [1,4]. According to recent studies, relapse can be observed in approximately 20% of children with ALL. Therefore, research on novel markers of individual predisposition and early detection of disease progression is pivotal for further improvement of disease-free survival in the leukemic subjects [19,20].

The microenvironment of the bone marrow, the primary niche of leukemia pathogenesis, has been under thorough investigation for years. That is especially related to increasing interest in the implementation of immunotherapy in ALL management [7,21]. To date, limited studies have reported on the significance of T-cell lymphocyte subsets (other than lymphoblasts) in patients with acute lymphoblastic leukemia, with only very few focusing on the pediatric population [11,22,23]. Extensive dissemination of ALL cells in the human body affects the proximity related constant interaction of lymphoblasts with immune cells. Thus, the imminent correlation between the patients’ imbalanced immune system and disease eradication is a fact and can be assessed thoroughly [24]. The role of CD4+CD25+FOXP3+ regulatory T cells (Tregs) in acute lymphoblastic leukemia development has been linked to a higher immunosuppressive activity compared to the Tregs of healthy subjects, leading to the inhibition of anti-tumor immunity [20,25]. The role of Th17 lymphocytes in cancer progression is yet to be unambiguously determined. While most scientists assign these lymphocyte subsets an antitumorigenic role, explained through their stimulation of immune cells into the cancer niche, several studies have now shed light upon their promoting role in tumorigenesis [26,27]. Nevertheless, the exact role of Th17 and Treg subsets in ALL has not been comprehensively studied to date, especially in the context of correlations with the clinical outcome in pediatric patients.

Considering the opposite roles of the discussed lymphocyte populations—effector Th17 promoting inflammation and mobilization of immune cells and Tregs counteracting excessive reactions—their increase can reflect efforts to maintain balance in the face of ongoing neoplastic process. Noteworthy, Th17 and Treg cells exhibit both tumor promoting and suppressive activity. IL-17 released by Th17 cells were found to induce angiogenesis and prevent cell apoptosis, thus, supporting survival of the tumor [28]. Here, we found that ALL patients demonstrated higher levels of Th17 and Treg lymphocytes in the bone marrow at the moment of diagnosis. Interestingly, despite those significant changes, such phenomenon was not observed in the peripheral blood, indicating the crucial contribution of the tested cells exclusively in the primary niche of the leukemic process. Actually, due to the most significant tumor-related phenomenon deriving from the bone marrow, these data do not seem surprising. Additionally, T-regulatory lymphocytes seem to have the tendency to accumulate in the bone marrow [29]. Interestingly, Salem et al. reported increased levels of Tregs in the peripheral blood of pediatric B-ALL patients as compared with healthy controls [30]. Xiang et al. reported, in their patients with B-cell acute lymphoblastic leukemia, higher levels of Treg and Th17 lymphocytes in the peripheral blood compared to healthy controls [11]. Whereas, Bi et al. observed such dependence in both peripheral blood and bone marrow niche [18]. Similar results to Bi et al. were achieved by Abousamra et al. in the context of both ALL and AML [24]. Whether this discrepancy is due to the concomitant incidence of an infectious process or the advancement of the neoplastic process is yet to be determined. Regardless of higher Treg and Th17 levels, we found that concentrations of IL-10 and IL-17 in bone marrow of the ALL patients were comparable to those of the healthy control group, at the time of diagnosis. At the same time, within leukemic subjects, a significant advantage of IL-10 over IL-17 was reported. This might suggest a suppressive role of leukemic cells in reference to Th17 cells, shifting Tregs towards intensified IL-10 production [31].

Stratification of the patients into groups based on Th17 or Treg levels allowed us to visualize the associations between those parameters and hematological results. We determined inter alia that in patients with lower levels of lymphoblasts in the peripheral blood significantly higher frequency of Treg was detected in the bone marrow. That might suggest that better control of the inflammatory processes in the specific tissue is required to achieve effective reduction in blasts. Noteworthy, our results are additionally supported by those of Niedzwiecki et al., where a negative correlation between lymphoblasts in the peripheral blood and regulatory T cells in the bone marrow was shown [29].

Monitoring the time-dependent change of IL-17-producing lymphocytes and regulatory T cells throughout treatment revealed a significant decrease in the latter population. Such reduction in the cells with immunosuppressive potential might be associated with the effectiveness in leukemia management and limitation of the related inflammatory process. Similar findings were reported in two other studies where a decline in FoxP3-positive cells was observed after the induction phase of ALL chemotherapy [25,26,30]. A decrease in regulatory T-cell population was also observed in patients who achieved clinical remission when compared to their initial level at the moment of diagnosis [25,32]. Interestingly, Idris et al. reported a normalization of the Treg levels after successful therapy, obtaining comparable results to those present in healthy individuals [33]. Noteworthy, this seems to be confirmed by the analogical time-dependent change of IL-10 and IL-17 levels in the bone marrow that seem to decrease upon treatment onset. Interestingly, therapy implementation gradually reduced dominance of the IL-10 concentration in ALL patients, thus, increasing influence of the IL-17. That seems to be consistent with the data of Brix et al., where a decline in IL-10 was noted with a subsequent increase in IL-17 levels upon chemotherapy [34]. Consistent decline in both Treg/Th17 and their related cytokines, IL-10/IL-17, might result from an exceptionally strong correlation between associated cytokines that we reported here.

Considering the information above, we hypothesized whether Th17 or Treg levels are related to the response of ALL patients to the induction phase of the treatment protocol. First, we found that those patients with higher Th17 cells exhibited a lower leukocyte count at the final time point of the patients’ monitoring (day 33 of the induction phase ALL-IC BFM 2009). At the same time, however, those subjects also demonstrated higher blasts levels at the moment of diagnosis. Regarding lymphoblasts presence, those differences were diminished in time and both groups managed to achieve an effective reduction in cancer cells within the bone marrow and peripheral blood. No significant changes were found in reference to changes in the level of regulatory T cells throughout ALL treatment. However, we observed a higher leukocyte count in children with lower regulatory T cells. On the contrary, Zahran et al. found a significantly higher level of CD4+CD25+high T-cells in pediatric ALL patients with leukocytes of 50 × 10^9^/L and higher compared to those with leukocytes below this threshold [32]. The effectiveness of therapy (based on a decrease in lymphoblast values) was comparable between low and high Th17/Treg. However, the balance between both subsets seems to have a crucial role in the course of ALL management. That is supported by the observation where a lower Th17/Treg ratio was associated with higher WBC counts at the last point of the monitoring.

Taking into account the significant role of Treg and Th17 balance in the pediatric acute lymphoblastic leukemia, we additionally focused on assessing their predictive value in monitoring the disease. Analysis of Th17/Treg balance was found to be essential in predicting the prevalence of the disease recurrence, where those with a lower ratio demonstrated a reduced risk for unfavorable prognosis. Considering the close relationship of the Th17 and Treg, monitoring their ratio in the peripheral blood was found to have clinical significance, especially in the context of cancer grade and patient survival. Although the imbalance between the level of Th17 and Treg lymphocytes can be determined in numerous cancer types, the ratio of Th17/Treg was found both to be elevated and depleted depending on the malignancy [35].

Even more crucial observations were reported in connection with MRD levels at 15th day of induction therapy. We managed to demonstrate that lower levels of Tregs are indeed related to worse outcome reflected by higher risk of high MRD prevalence. These results are in concordance with a study where an evaluation of Tregs in the course of treatment of pediatric patients with acute lymphoblastic leukemia exhibited a low negative correlation between MRD level and regulatory T-cells in the bone marrow [29]. Noteworthy, shorter overall survival and relapse free survival was associated with higher levels of circulating Tregs [36]. In contrast, a more beneficial outcome was related to lower frequencies of Th17 cells and their ratio when combined with Tregs. Interestingly, on the contrary, according to the data presented by Abousamara et al., patients with higher circulating Th17 level can be associated with a longer median overall survival [24].

Here we provide, to our knowledge, the first such thorough correlation of Th17 and Treg immune cell subset population with the clinical aspect of childhood acute lymphoblastic leukemia in a homogenous patient group. However, we are aware of the limitations of our work which can be improved in the future, predominantly through implementation of multicenter studies allowing for a higher number of subjects. Even though acute lymphoblastic leukemia is the most commonly occurring childhood cancer, it is still considered a rare disease, which makes it difficult to collect a relatively large study group.

## 4. Materials and Methods

### 4.1. Patients’ Description

Pediatric patients diagnosed with acute lymphoblastic leukemia were enrolled in the study upon admission to the Department of Pediatric Oncology and Hematology, Medical University of Bialystok, Poland during the years 2013–2017. A total number of 35 patients and 48 healthy control subjects were involved in the experimental procedures described below. The healthy control group consisted of patients selected during routine diagnostic procedures, with excluded oncological, autoimmune or other inflammatory conditions. Study groups were consistent in terms of age (*p* = 0.0666) and sex distribution (*p* = 0.1631). Biological material used in the experiments was bone marrow and peripheral blood collected during standard diagnostic procedures conducted at our clinical department. Informed consent was obtained from all patients (parent or legal guardian for underage subjects). The study protocol was approved by the Ethical Committee at the Medical University of Bialystok—approval number—APK.002.603.2021. All patients were treated in accordance with the ALL-IC BFM 2009 Protocol. The most significant details of the studied groups’ characteristics were presented in Table 1. The bone marrow and peripheral blood were collected during Induction Therapy (days 1, 8,15 and 33) and before Protocol M/HR-1. MRD (minimal residual disease) was assessed by flow cytometry on day 15 and was one of the factors of risk-group stratification. The Induction Therapy (Protocol I’A/IA) includes the use of prednisone, vincristine, daunorubicin, L-asparaginase and intrathecal methotrexate.

### 4.2. Flow Cytometric Assessment of Th17 and Treg Lymphocytes

Whole bone marrow or whole peripheral blood was subjected to immunostaining with fluorochrome-conjugated monoclonal antibodies, using 200 μL of material for each sample. The panel of antibodies implemented in the ex vivo analysis included: anti-CD4 FITC (clone RPA-T4), anti-Foxp3 PE (clone 259D/C7), and anti-IL-17A PE (clone SCPL1362) (BD Bioscience, San Jose, CA, USA). For the assessment of Th17 lymphocytes, samples were incubated for 5 h at 37 °C, Leukocyte Activation Cocktail with brefeldin A (BD Pharmingen, San Diego, CA, USA), to increase efficiency of the intracellular cytokine detection. Extracellular staining of the CD4 marker and washing out of unbound antibodies after incubation was followed by permeabilization with Permeabilization Buffer 2 (BD Bioscience). Subsequently to intracellular staining with anti-Foxp3 and anti-IL-17A antibodies, incubation, washing steps, cells fixation was performed with the use of CellFix reagent (BD Bioscience), and samples were stored at 4 °C. Data were acquired the same day on FACS Calibur flow cytometer (BD Bioscience, San Jose, CA, USA). FlowJo software 10.7.2 (Tree Star Inc., Ashland, OR, USA) was applied for the processing of the flow cytometric data. Morphology using relative size (forward scatter, FSC) and granularity/complexity (side scatter, SSC) properties, was the initial parameter to distinguish lymphocytes. The CD4 marker allowed for selection of Th cells within lymphocytes. Furthermore, frequency of regulatory T (Treg) and IL-17-producing T cells (Th17) was assessed on the basis of Foxp3+ and IL-17A expression, respectively. Gating strategy implemented in the study, including proper controls applied (unstained, FMO and isotype control), has been presented on Figure 1A.

### 4.3. Immunoenzymatic Assessment of IL-10 and IL-17 in the Bone Marrow

Bone marrow supernatants of ALL patients and the healthy control group were subjected to immunoenzymatic (ELISA) assessment of cytokines related to Treg and Th17 activity, namely IL-10 and IL-17, respectively. Implemented protocols included those provided by the manufacturer of the DuoSet ELISA Kits (R&D system, Minneapolis, MN, USA). Limits of detection for tested cytokines were 31.2–2000 pg/mL for IL-10 and 15.6–1000 pg/mL for IL-17. Cytokines’ concentration-related absorbance level was measured at 450 nm wavelength with the use LEDETECT96 microplate reader (Labexim Products, Lengau, Austria). A standard curve (four-parameter logistic (4-PL) curve-fit) was used for final establishment of the tested proteins’ concentrations. All the samples were analyzed in duplicates.

### 4.4. Biostatistical Analysis

Statistical processing of the collected data was performed using GraphPad Prism 9.0 statistical software (GraphPad Prism Inc., San Diego, CA, USA). A non-parametric Mann-Whitney U test was implemented, due to non-Gaussian distribution of the data, to compare differences between studied groups. An ANOVA test was used to evaluate changes between different time points in the course of therapy (with Fisher’s LSD post-test). Significant differences were considered for those with *p* value of below 0.05 and indicated with asterisks or exact *p* value on the graphs: *—*p* < 0.05, **—*p* < 0.01, ***—*p* < 0.001, ****—*p* < 0.0001. Graphical presentation of the results included median values, interquartile range (25th and 75th quartile), and additionally minimum/maximum values when Turkey boxes were used. Considering data distribution, the parametric Spearman correlation test was used for determination of associations between selected parameters. Exact coefficient values (R value) and significance indicated with asterisks were presented on the graphs. In the relative risk assessment, Fisher’s exact test was applied, and the results were presented as relative risk values (with 95% confidence interval) together with *p* value.

## 5. Conclusions

In summary, we demonstrated a significant role of Th17 and Treg lymphocytes in acute lymphoblastic leukemia of pediatric patients and their contribution to the disease-related parameters. Importantly, we revealed how significant is the immune balance in the course of a neoplastic process as far as effector Th17 and regulatory T cells are considered. We showed that better understanding of these immune populations might give the possibility to influence the course of applied ALL treatment. In fact, to our knowledge, we demonstrated for the first time the value of Th17 and Treg lymphocytes simultaneous assessment in predicting the outcome of acute lymphoblastic leukemia therapy in the pediatric patients’ population. We believe that further studies would allow for determination of mechanisms constituting the background of the observed phenomenon. Moreover, clinical implementation of the Th17 and Treg predictive potential at an extended scale is required to validate their routine use in the course of ALL treatment monitoring.

## Figures and Tables

**Figure 1 ijms-24-12323-f001:**
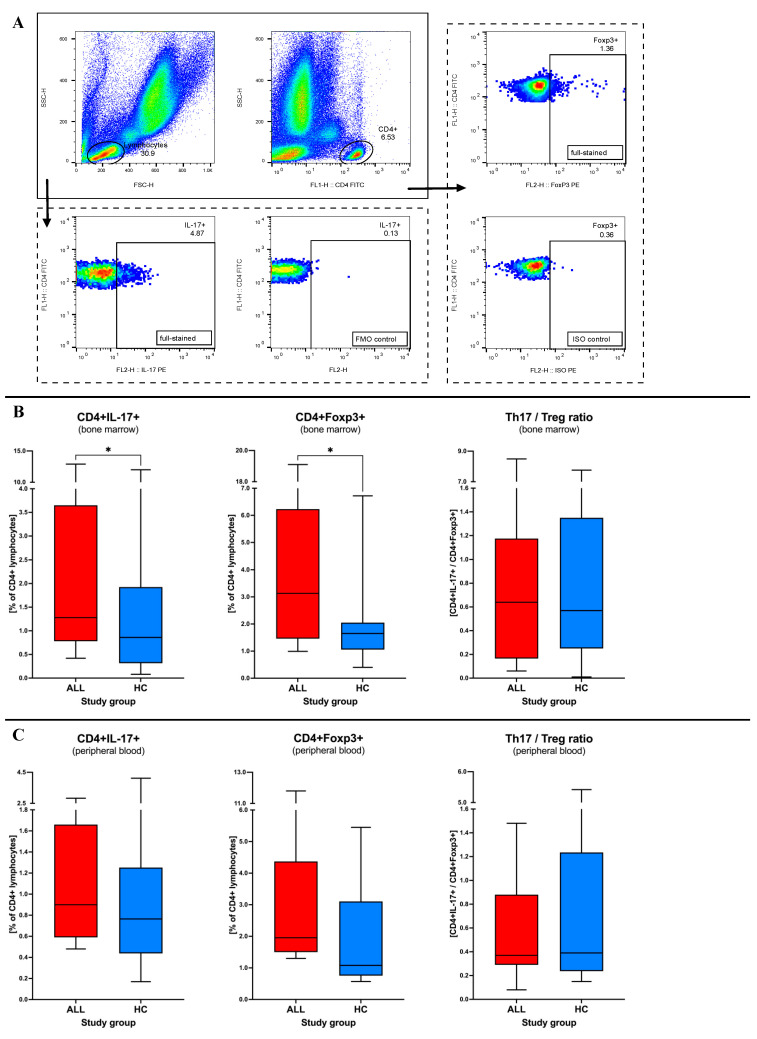
Changes in Th17 and Treg lymphocytes in patients with acute lymphoblastic leukemia. Demonstration of gating strategy used for detection and analysis of IL-17-producing T cells and regulatory with expression of Foxp3 (specific controls used included) (**A**). Initial Th17 and Treg cells levels were assessed before therapy within bone marrow (**B**) and peripheral blood (**C**) of leukemia patients (ALL) at the time of initial diagnosis and healthy control group (HC). Data presented as medians and interquartile range (boxes), with minimum and maximum indicated with whiskers. Data presented as Turkey boxplots indicating median values and data distribution (minimum and maximum). Statistically significant values were indicated with asterisks: *—*p* < 0.05.

**Figure 2 ijms-24-12323-f002:**
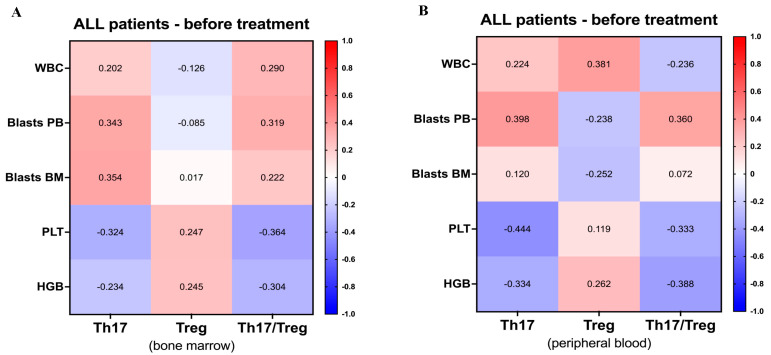
Correlations between Th17 and Treg lymphocytes and laboratory diagnostic parameters. Mutual associations between studied immune cells from bone marrow (**A**) or peripheral blood (**B**) and laboratory results were evaluated. Data presented as r values and statistical significance indicated with asterisks.

**Figure 3 ijms-24-12323-f003:**
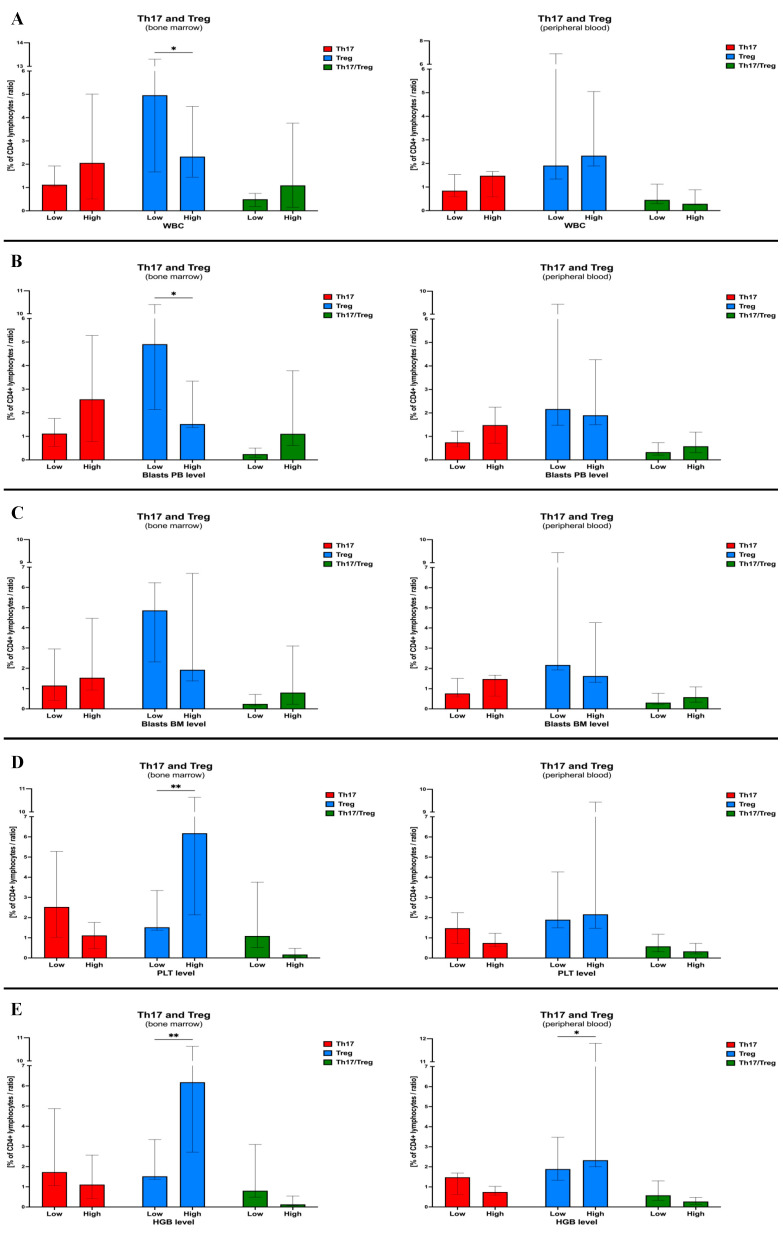
Relation between disease-related parameters and Th17 or Treg lymphocyte levels in acute lymphoblastic leukemia. Differences in Th17 and Treg cells were evaluated in ALL pediatric patients stratified into median-based groups with low/high values of white blood cell count (WBC) (**A**), blast frequency in peripheral blood (PB) (**B**) and bone marrow (BM) (**C**), platelets level (PLT) (**D**), and hemoglobin concentration (HGB) (**E**). Data presented as median values and interquartile range. Statistically significant values were indicated with asterisks: *—*p* < 0.05, **—*p* < 0.01.

**Figure 4 ijms-24-12323-f004:**
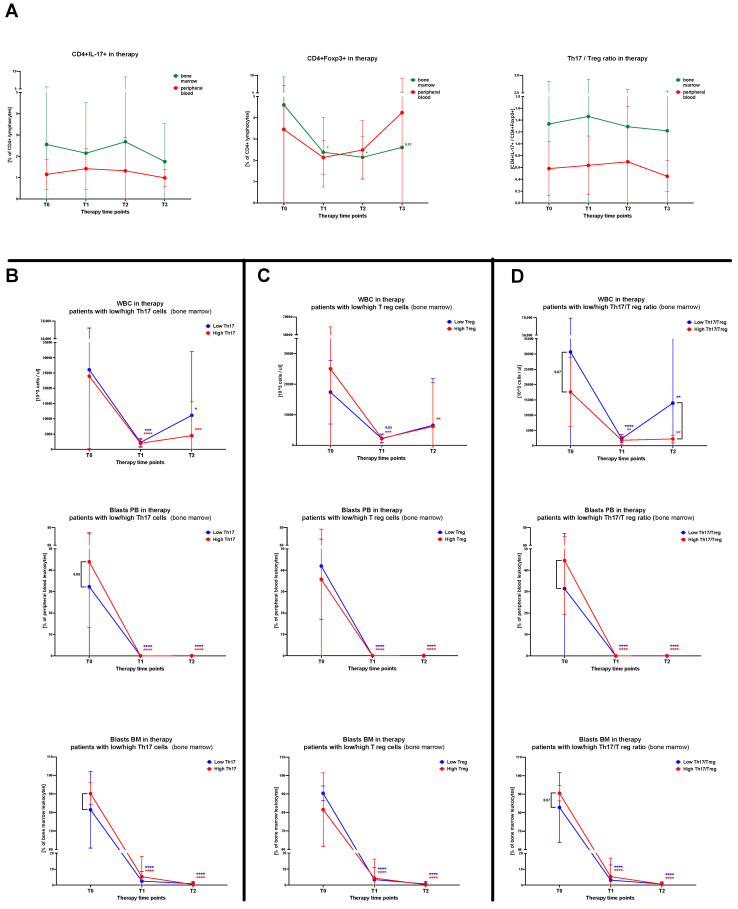
Therapy-related variations in Th17 and Treg lymphocyte levels of acute lymphoblastic leukemia patients. Peripheral blood and bone marrow IL-17-producing and Foxp3-expressing T cells were monitored in the course of therapy (**A**). The influence of tested immune cell levels on therapy-induced changes in white blood cell count (WBC), frequency of blasts in peripheral blood and bone marrow was analyzed in context of Th17 (**B**) and Treg (**C**) lymphocyte as well as Th17/Treg ratio (**D**) levels (median-based low/high level stratification). Data presented as median values and interquartile range. Statistically significant values were indicated with asterisks: *—*p* < 0.05, **—*p* < 0.01, ***—*p* < 0.001, ****—*p* < 0.0001.

**Figure 5 ijms-24-12323-f005:**
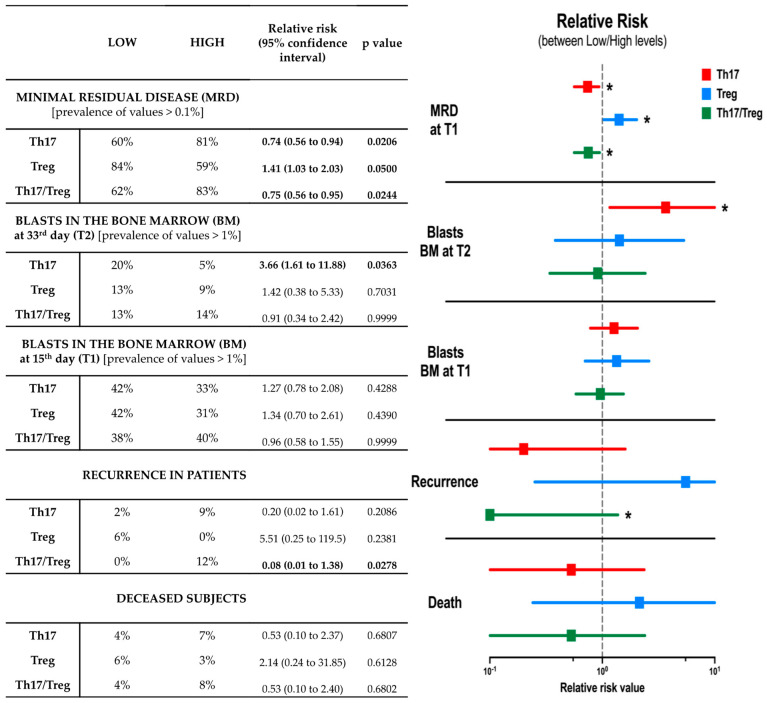
Evaluation of prognostic values of initial (pre-therapy) Th17 and Treg levels in assessing the acute lymphoblastic leukemia treatment outcome in pediatric patients. Tabular data demonstrating frequencies of ALL-related parameters prevalence within studied patients, together with relative risk values (with 95% confidence interval) and *p* value of analyzed differences. Data supported by visual presentation of relative risk values of selected parameters occurrence between patients with initially low versus high levels of Th17 and Treg cells, and Th17/Treg ratio. Statistically significant values were indicated with asterisks: *—*p* < 0.05.

**Figure 6 ijms-24-12323-f006:**
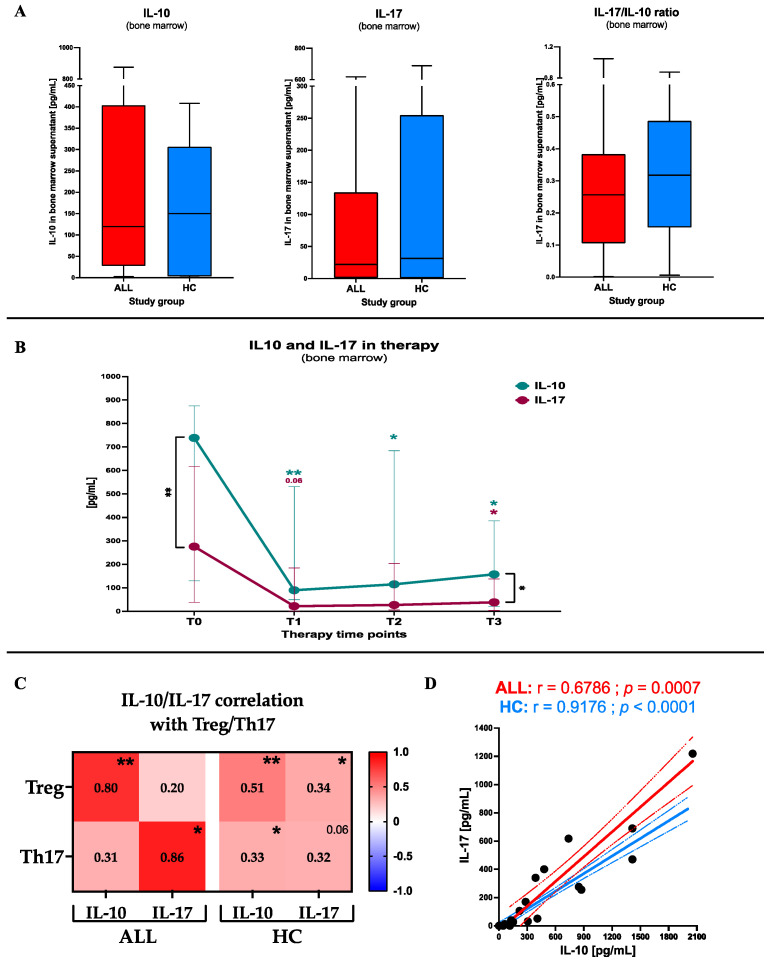
Assessment of IL-10 and IL-17 in the bone marrow of ALL patients. Initial differences in IL-10, IL-17, and their ratio, between ALL subjects and the healthy control group (data presented as medians and interquartile range (boxes), with minimum and maximum indicated with whiskers) (**A**). Therapy induced changes in the cytokines tested in leukemic patients (data presented as median values and interquartile range) (**B**). Mutual correlations between Treg or Th17 and selected cytokines (IL-17 and IL-10) (data presented as r values and statistical significance indicated with asterisks) (**C**). Comparative presentation of differences between ALL patients and control group in context of mutual correlations between IL-10 and IL-17 (**D**). Statistically significant values were indicated with asterisks: *—*p* < 0.05, **—*p* < 0.01.

**Table 1 ijms-24-12323-t001:** Characterization of the studied groups. Basic subject numbers, age and sex distribution, together with laboratory description of ALL patients—before and in the course of therapy. Data presented as mean values with standard deviation.

Parameter		All Patients (All)	Healthy Control Group (HC)
**Subjects** **Number**		35	48
**AGE** [years]		6.77 ±4.75	9.23 ±5.25
**Sex Distribution** [%male/%female]		66%/34%	52%/48%
**All Patients Laboratory Data**			
**Parameter**	**Before** **Therapy** **(T0)**	**15th Day** **of Therapy** **(T1)**	**33rd Day** **of Therapy** **(T2)**
**White Blood Cell** **(WBC)** [10^3^ cells/μL]	32.15 ±45.28	2.14 ±1.30	5.98 ±13.45
**Blasts In Peripheral Blood (PB)** [% of leukocytes]	42.91 ±34.37	0.00 ±0.00	0.06 ±0.24
**Blasts In Bone Marrow (BM)** [% of leukocytes]	83.63 ±14.65	4.53 ±11.11	0.10 ±0.30
**Platelets** **(PLT)** [10^3^/μL]	100.00 ±104.89		
**Hemoglobin** **(HGB)** [g/dL]	7.78 ±2.04		

## Data Availability

Data is contained within the article. The data presented in this study are available at the Department of Regenerative Medicine and Immune Regulation, Medical Univeristy of Bialystok, Poland upon request.

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
