# Peer review of "Significance of Th17 and Treg in Treatment Efficacy and Outcome in Pediatric Acute Lymphoblastic Leukemia"

_ijms, 2023, doi:10.3390/ijms241512323_

Round 1

Reviewer 1 Report

This manuscript addresses a topic that can be of interest to readers. It is clearly organized and based on good hypotheses. The work is convincing although some minor points need to be discussed.

1. Flow cytometric detection of Th17 and Treg: What was used for assessment? Mononuclear cells (If so, how were they isolated?), whole blood or whole bone marrow?

2. Flow cytometry: There is no mention of important details such as how the flow cytometer was set up, quality control and compensation. How was the data verified? In particular, intracellular staining by flow cytometry is difficult and can often lead to false positives. Authors should provide a variety of controls (ie. cells unstained, isotype control, etc). How many cells were processed?

3. Mention, whether the ELISA assays were carried out in duplicates for each sample.

4. Results: Authors should better describe the analysis performed, the number of cases/samples analyzed for each parameter, and the statistical analysis results (median, range, IQR...).

5. Figure 3.: The authors divided the cohort into two groups: low and high. What were the cutoff values? Some explanation is necessary in the main text.

6. All abbreviations should be defined.

7. There are a few grammatical and typo errors which must be corrected.

Reviewer 2 Report

A nice, interesting, and well-written artcile, exploring the role of a few inflammatory chemokines and the related cells in ALL.

I have only a few minor observation:

1) Lane 291: frequencies...of what? please rephrase as not very clear for the reader

2) The levels of cytokines and cell populations were evaluated along the disease course. It would be nice to see a comparison between, patients achieving CR vs. no; deep CR (no MRD) vs no CR ; and then see whether patients in CR but still higher values did relapse more frequently (relapse rate in Higher vs lower levels/frequencies)
